# JWST interferometric imaging reveals the dusty torus obscuring the supermassive black hole of Circinus galaxy

Enrique Lopez-Rodriguez [1,2] ✉, Joel Sanchez-Bermudez [3], Omaira González-Martín [4], Robert Nikutta [5], Ryan M. Lau [5], Deepashri Thatte [6], Ismael García-Bernete [7], Julien H. Girard [6] & Matthew J. Hankins [8]

The dusty and molecular torus is an elusive structure surrounding supermassive black holes, yet its importance is unequivocal for understanding feedback and accretion mechanisms. The torus and accretion disk feed the inspiraling gas onto the active nucleus, launching outflows that fundamentally connect the active nucleus's activity to the host galaxy. In this work, we utilize the aperture-masking interferometric mode onboard the JWST to achieve a resolution of 0.08" at 4.3 $\mu$m and bring out the fainter features in the central 10 pc of the Circinus galaxy. We show that most of the dust mass is located along the equatorial axis in the form of a 5 × 3 pc disk feeding the active nucleus. Only < 1% of the dust emission arises from an arc structure composed of hot dust entrained in a molecular and ionized outflow, while the extended emission is associated with dust heated by the active galaxy at large scales.

The buildup of the central supermassive black hole (SMBH) mass is a fundamental facet of galaxy growth and evolution and occurs at least in part through active galactic nuclei (AGN) accretion in massive galaxies. On the other hand, AGN feedback via wide-angle winds can quench[1] and/or enhance[2] star formation in galaxies[1], regulate SMBH accretion[3], and even shape galaxy morphology[4]. The characterization of the central 1–100 pc around AGN reveals the origins of accretion and feedback mechanisms during galaxy evolution and provides constraints for theoretical models of galaxy formation[5].

The bulk of dust and molecular mass cospatial with accreting and outflowing material is studied at sub-mm wavelengths using ALMA [Atacama Large Millimeter/submillimeter Array;[6]]. Further mid-infrared (MIR; 7–12 $\mu$m) interferometric observations are more sensitive to the extended emission above and below the disk. This MIR emission predominantly arises from warm (200–400 K), optically thin dust layers in the torus walls and/or a dusty wind[7,8] launched by radiation pressure from a magnetohydrodynamical wind generated at sub-pc scales[9–12].

The near-IR (NIR; 1–5 $\mu$m) observations are sensitive to the hot dust (400–1500 K), which traces the inner edge of the torus and/or the base of the torus walls or dusty winds[7,13,14]. Surprisingly, the physical structures that produce the measured 3–5 $\mu$m excess emission in AGN since first observed in the early '90s still remains unclear[15–20]. This observed NIR excess is thought to arise either from hot dusty winds, hot graphite dust in the inner torus, and/or residual starlight from the host galaxy. Unequivocally, any of these scenarios has critical consequences on the accretion and feedback mechanisms on the buildup of the central SMBH. Thus, the identification of the physical structure and mechanism producing the NIR excess will allow us to connect the

[1]Department of Physics & Astronomy, University of South Carolina, Columbia, SC, USA. [2]Kavli Institute for Particle Astrophysics & Cosmology (KIPAC), Stanford University, Stanford, CA, USA. [3]Instituto de Astronomía, Universidad Nacional Autónoma de México, Ciudad de Mexico, Mexico. [4]Instituto de Radioastronononomía y Astrofísica (IRyA), Universidad Nacional Autónoma de México, Morelia, Michoacán, Mexico. [5]NSF NOIRLab, Tucson, AZ, USA. [6]Space Telescope Science Institute, Baltimore, MD, USA. [7]Centro de Astrobiología (CAB), CSIC-INTA, Villanueva de la Cañada, Madrid, Spain. [8]Arkansas Tech University, Russellville, AR, USA. ✉e-mail: elopezrodriguez@sc.edu

accretion disk with the reservoir of gas feeding it and the interaction with the host galaxy.

The Circinus galaxy is the best candidate to solve a major issue in AGN physics within the central 10 pc: wind (outflow) vs. torus (accretion). Circinus hosts the nearest, $4.2 \pm 0.7$ Mpc [0.1" = 2 pc;[21]], type 2 Seyfert galaxy with an intermediate bolometric luminosity of $L_{bol} = 10^{43.6}$ erg s$^{-1}$ [22–24], and shows well-defined inflowing and outflowing dusty and molecular material from the host spiral galaxy and its AGN[25]. It has a kpc-scale radio jet at a position angle, PA about −64° [26] and a pc-scale radio jet at a PA about −84° [25].

In the NIR, a $30 \times 4$ mas$^2$ ($0.6 \times 0.08$ pc$^2$) disk-like structure at a PA about 7° East of North dominates the emission at $4.7\,\mu m$ (4 mas resolution) using observations with the Multi-aperture Mid-Infrared Spectroscopic Experiment (MATISSE) on the Very Large Telescope Interferometer (VLTI)[27]. However, this disk structure only accounts for <5% of the measured flux within the 0.4" aperture measured by the Nasmyth Adaptive Optics System-Coude Near Infrared Camera (NACO) on the VLT in the L-band ($4.8\,\mu m$)[28]. Thus, the origin of the NIR extended emission within 0.1–0.5" (2–10 pc), is still missing, and the total emission of directly heated dust in the torus and/or winds remains unknown.

In the MIR, Circinus has a compact 1.9 pc diameter central disk and 5 pc, in diameter, extended diffuse dusty component perpendicular to the maser disk using N-band (8–13 $\mu m$; 9 mas resolution) MATISSE/VLTI observations[29]. This MIR emission extends up to approximately 20 pc on only one side of the ionization cone walls as observed using the VLT spectrometer and imager for the mid-infrared (VISIR)[30]. The central 5 pc-scale dusty emission extension has a wider opening angle and is clumpier than the 20 pc-scale structure. The N-band observations have been characterized using a series of torus-only and torus+winds models, concluding that the most likely scenario is given by a clumpy disk+hyperboloid component[30]. Under this scenario, the warm (200–400 K) dusty extended emission is thought to arise from anisotropic radiation at the location of the warped maser disk[31]. The 1–0 $\mu m$ spectral energy distribution (SED) shows that the N-band observations strongly drive the model fit. However, models are highly degenerate when explaining the hot (>400 K) dust component at NIR wavelengths.

In this work, we quantify the morphology and properties of the hot dust in the torus and extended structures, and identify the dominant physical structure responsible for the hot and warm dusty extended emission within the central 10 pc of Circinus. Combined with continuum data, gas tracers, and torus models, we show that most of the dust mass is located in the equatorial axis in the form of a disk feeding the AGN. Only <1% of the total dust emission is located in a hot and warm outflow, while most of the extended emission is dust radiated by the active nucleus.

## Results

### New JWST interferometric observations and images

We observed the Circinus galaxy in July 2024 and March 2025 with the Aperture Masking Interferometry [AMI;[32]] mode in JWST's Near Infrared Imager and Slitless Spectrograph [NIRISS;[33]] at 3.8 $\mu m$ (F380M), 4.3 $\mu m$ (F430M), and 4.8 $\mu m$ (F480M) (Methods section 'Observations'). Both observations ensure a 90° rotation of the uv-plane to increase its coverage, minimizing image reconstruction artifacts. The 65 mas NIRISS pixels are Nyquist sampled at 4 $\mu m$ in the medium band. Interferometric observables, closure phases and square visibilities[34,35], are extracted from the calibrated data and used for image reconstruction. We used SQUEEZE[36] for the image reconstruction. Additionally, we performed a bootstrapping analysis on the uv-plane to obtain the significance level of the features in our final images (Methods section 'Image reconstruction'). We used the peak emission from the interferogram images to assign the world coordinate system in the reconstructed images (Methods section 'WCS correction'). A standard star with known IR fluxes, previously used to perform the flux calibration of the MATISSE/VLTI observations of Circinus[27,29], was observed after the Circinus observations and used to perform the flux calibration at each filter (Methods section 'Flux calibration'). We estimate that emission lines have small contributions, <10%, within the AMI filters (Methods section 'Emission line contribution'). Figure 1 shows the SQUEEZE final reconstructed images of Circinus with angular resolutions ($\lambda/2B$, where $\lambda$ is the wavelength and $B$ is the baseline of 6.5 m) of $93 \times 88$ mas$^2$ ($1.9 \times 1.8$ pc$^2$), $105 \times 101$ mas$^2$ ($2.1 \times 2.0$ pc$^2$), and $123 \times 116$ mas$^2$ ($2.5 \times 2.3$ pc$^2$) in the F380M, F430M, and F480M filters, respectively.

At all wavelengths (3.8–4.8 $\mu m$), we find that the continuum dust emission has 1) an extended component of $0.10" \times 0.25"$ ($2 \times 5$ pc$^2$) at a PA = −70°, b) a 'North arc' feature extending 4 pc toward the north-east direction, and c) a lack of IR emission ('Holes') in the north-east and south-west regions at 0.4" (8 pc) at a PA 50° from the nucleus. The PA of the extended emission is estimated as the angle along the long axis of the extension fitted by a 2D Gaussian, and the PA of the 'Holes' is taken from the long axis of the HCN(3-2) image shown in Fig. 2. We identify the 'North arc' as a real feature with > 16$\sigma$ detection in the reconstructed images of all filters, as it persists after our bootstrapping analysis and independent image reconstructions (Methods section 'Image reconstruction'). In addition, all images show an extended low-surface brightness emission at > 5 pc at a 4$\sigma$ significance, but it only accounts for < 1% of the total flux within the field-of-view ($14 \times 14$ pc$^2$). The low-surface brightness emission increases from southeast to northwest, and it is mostly along the directions of the narrow line region (NLR). The obscuration from the host galaxy may produce the variation in flux; the southern area is behind the galaxy disk, and the northern region is above it[37]. In addition, this low-surface brightness emission also forms arcs around the 'Holes'.

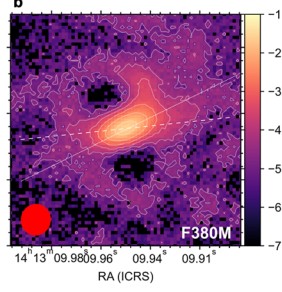
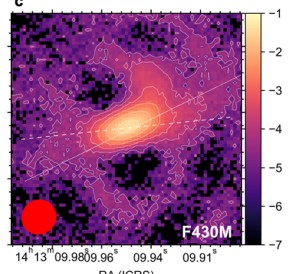
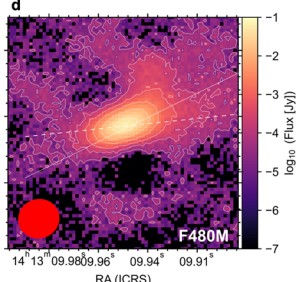

**Fig. 1 | The dust emission of the central $14 \times 14$ pc$^2$ of the Circinus galaxy observed with AMI/JWST. a** The RGB image (Red: F480M, Green: F430M, Blue: F380M) of Circinus with the orientations of the kpc-scale [white dotted line, PA = − 64°;[26]], pc-scale [white dashed line, PA = − 84°;[25], see Fig. 2] radio jets, the `North Arc', and `Holes' features. A 2 pc scale is shown. **b–d** The AMI/JWST observations of the continuum dust emission at 3.8 $\mu m$ (F380M), 4.3 $\mu m$ (F430M), and 4.8 $\mu m$ (F480M). The beam sizes (red ellipses) of $93 \times 88$ mas$^2$ ($1.9 \times 1.8$ pc$^2$), $105 \times 101$ mas$^2$ ($2.1 \times 2.0$ pc$^2$), and $123 \times 116$ mas$^2$ ($2.5 \times 2.3$ pc$^2$) are shown in each panel. The contours start at 4$\sigma$ and increase in steps of $2^n\sigma$ with $n = 2, 4, 6, \ldots$. Dec. declination, RA right ascension.

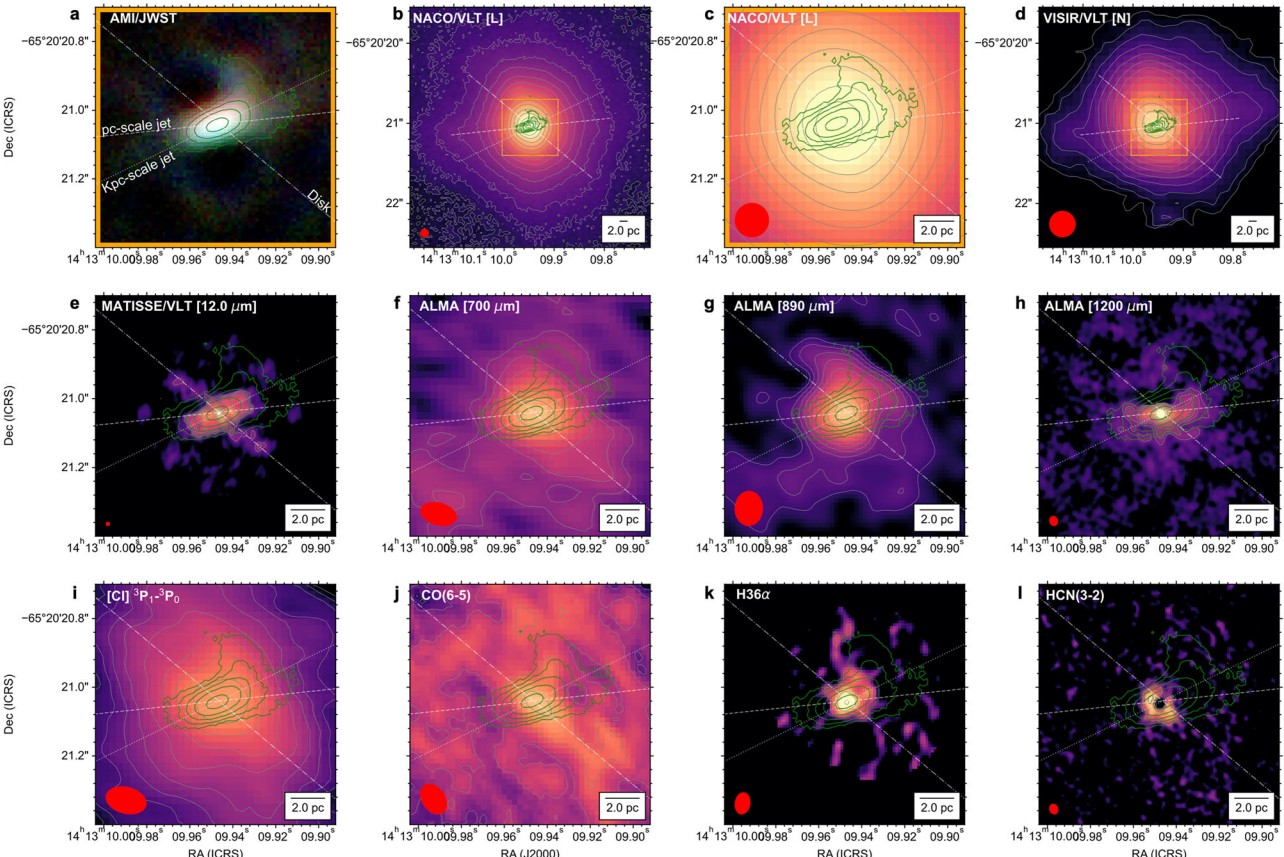

**Fig. 2 | The multi-phases of the Circinus galaxy. a** The RGB image of Circinus using the AMI/JWST observations and green contours of the 4.3 μm AMI/JWST observations as shown in Fig. 1. The orientations of the pc-scale jet (white dashed line), kpc-scale jet (white dashed line), and disk (white dotted-dashed line) are shown. **b–d** The single-dish continuum observations at 4.3 μm (L-band) and 10 μm (N-band) with the VLT[28] in the central 60 × 60 pc². The orange rectangle shows the 14 × 14 pc² within the 60 × 60 pc² large FOV images. **e–h** The interferometric continuum observations at 12.0 μm with MATISSE/VLTI[29] and at 700, 890, and 1200 μm with ALMA[25]. **i–l** The interferometric observations[25,40] of several gas tracers: [CI] $^3P_1$-$^0P_1$, CO(6-5), H36α, and HCN(3-2). The gray contours show the morphology of the color scale images in each panel. The beam of each observation (red ellipse) and a 2 pc legend are shown.

The AMI/JWST observations improve the angular resolution of NIRCam/JWST[38] by a factor of two−0.08″ vs. 0.14″ at 4.3 μm−, while removing the typical shape of the point spread function (PSF) of the JWST and filtering out the large-scale starlight emission from the host galaxy (the maximum recoverable scale is 0.5″ at 4.8 μm). These benefits are clearly visualized in Fig. 2-top. Note that NIRCam/JWST observations will produce saturated images for Circinus, and that the 8-m class single-dish observations using NACO/VLT and VISIR/VLT observations at 4.8 μm[28] and 10.5 μm[30], respectively, are dominated by the large-scale starlight emission. Our AMI/JWST observations of Circinus show that the NIR emission within the central 10 pc of Circinus is dominated by an extended emission feature physically linked to the AGN.

**The multi-phase components of the extended emission**

We analyze the spatial correspondence of our measured 3.8–4.8 μm extended emission features with those arising from thermal and non-thermal continuum emission and molecular and ionized gas tracers (Methods section 'Archival Observations'). We use observations with comparable, or better, angular resolution than the AMI/JWST observations. From the continuum emission observations (Fig. 2e), the 3.8–4.8 μm emission is cospatial, in extension and orientation, with the 8−13 μm (N-band) emission observed with MATISSE/VLTI[29]. The 8–13 μm extended emission has a dust temperature of 200–240 K along the direction of the kpc-scale jet, while it reaches a maximum dust temperature of about 270 K in the two East and West knots from the core and along the direction of the pc-scale jet. In addition, the MATISSE/

VLTI observations[27] at 3.7 μm (L-band) show an unresolved source (0.3 pc in diameter) and a resolved structure, 0.3 × 0.6 pc², at 57° at 4.7 μm (M-band), highly offset from our extended emission (PA = − 70°). Due to the maximum recoverable angular scale of 80 mas in L-band and 360 mas in N-band of the MATISSE/VLTI observations, the 'North arc' is filtered out.

The 700–1200 μm continuum observations using ALMA (Fig. 2f–h) show variations in morphology due to the change in the physical mechanism producing the emission[25]. The 700 μm observation is dominated by thermal dust continuum emission with an extension of 4 × 10 pc² at a PA = 50° (Methods Section 'SED' and Supplementary Fig. 6). This structure is cospatial with the equatorial axis (i.e., disk) of the torus in Circinus and highly offset, ΔPA = 60°, from our measured 3.8–4.8 μm extended structure. The 700 μm dust continuum emission along the disk fills the lack of emission ('Holes') observed in our 3.8–4.8 μm AMI/JWST observations. Note also that the thickness, 4 pc, of the 700 μm dust emission spatially coincides with the long axis of the 3.8–4.8 μm extended emission. The 1200 μm observation is dominated by non-thermal synchrotron emission arising from an unresolved nucleus (0.5 pc), and a pc-scale jet extending 4 pc in diameter at a PA = −84° in the east-west direction. The eastern region of the synchrotron emission is cospatial with the dust continuum emission of our 3.8 −4.8 μm AMI/JWST observations. In the western and eastern regions, the synchrotron emission shows arcs toward the north direction at 2 and 1 pc from the nucleus, respectively. The western arc is cospatial with the beginning of the 'North arc' from our 3.8–4.8 μm AMI/JWST observations. The two hottest knots within

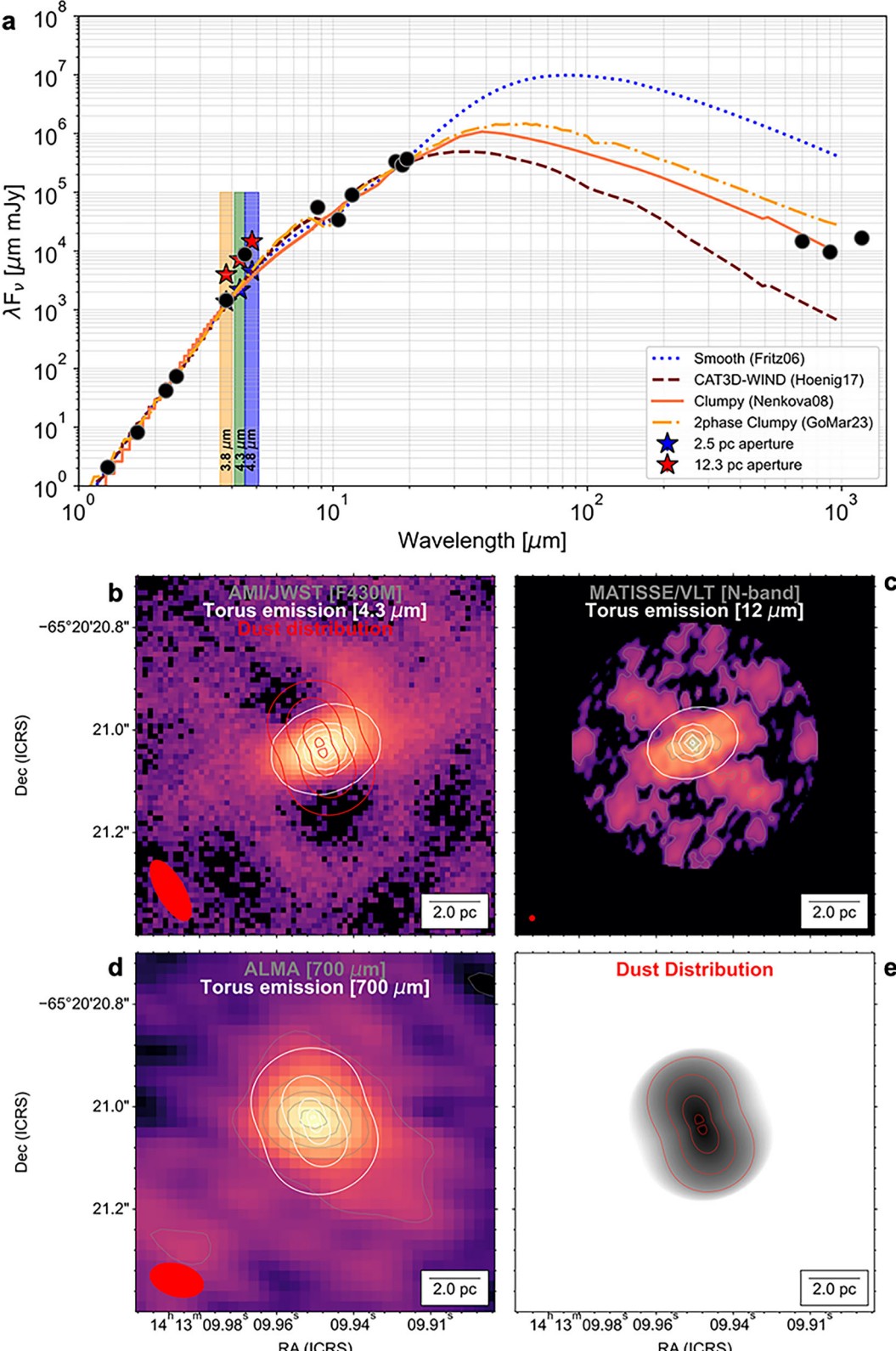

**Fig. 3 | The clumpy torus model describes the 1–1000 μm SED and morphological changes of the dust emission of Circinus. a** The 1–1000 μm photometric measurements from the literature (black dots) and our AMI/JWST observations at a 2.5 pc aperture (blue star) and at a large aperture of 12.3 pc (red star) of the central emission. The bandwidths of the AMI/JWST filters are shown as shadowed regions.

The best-fits of the several torus models are shown. Labels indicate the dominant emitting and physical components. **b**–**d** Dust emission at 4.3 μm, 10 μm, and 700 μm as shown in Fig. 2 with the synthetic dust emission (white contours) using HYPERCAT with the best-fit CLUMPY torus model. The dust mass distribution of the best-fit CLUMPY torus model is shown in red contours and **e**.

the central 0.5 pc measured in the N-band with MATISSE/VLT[29] are cospatial with the pc-scale jet at 1200 µm. The 890 µm observations show the combined emission from thermal dust and non-thermal synchrotron (Methods section 'SED'). This image has an unresolved core of 100 mas (2 pc), two arcs toward the north-east and south-west directions, and low surface brightness in the south-western region cospatial with the min-spiral arm[39,40]. The 890 µm northern arc is slightly offset toward the inner side of the 'North arc' at 3.8–4.8µm. In addition, the 890 µm image also shows a lack of emission along the northeast region of the disk at PA = 50°. This lack of emission is spatially coincident with the 'Holes' from the AMI observations, indicating a dust-dominated region.

From the continuum observations, we conclude that our measured $2 \times 5$ pc² extended emission is highly offset, $\Delta PA = 60°$, from the equatorial axis of the obscuring disk around the nucleus with an east-west component parallel to the pc-scale radio jet at a PA = −84° extending > 5 pc from the nucleus. This extended emission is also observed at 10 µm at 10s parsecs (Fig. 2d), which was attributed to the dusty cone edge directly radiated by the anisotropic radiation from the AGN[30]. The anisotropic radiation from the AGN may arise from the sub-pc warped maser disk[31]. In addition, there is a lack of 3.8–4.8 µm dust emission in our AMI/JWST observations, where the equatorial axis of the obscuring disk is present at 700 µm.

From the gas tracer observations (Fig. 2-bottom), the [CI] $^3P_1$-$^0P_1$ and H36α show two arcs along the northwest direction, which have previously been identified as multi-phase gas outflows[25]. The [CI] $^3P_1$-$^0P_1$ molecular image shows an outflow towards the northwest region from the core with two arcs towards the north and south at 2 pc scales. The North molecular [CI] $^3P_1$-$^0P_1$ outflow is spatially coincidental with the 'North arc' measured in our 3.8–4.8 µm AMI/JWST observations. The H36α traces the ionized outflow within the central pc and shows the same two arcs as in the molecular outflow with an unresolved (0 mas; 0.12 pc) core. The pc-scale ionized H36α outflow is spatially coincidental with the base of the 'North arc' in the 3.8–4.8 µm AMI/JWST observations. The CO(6-5) observations also show an emission structure cospatial with the first 2 pc of the 'North arc' detected in the 3.8–4.8 µm wavelength range.

The HCN(3-2) observations shows an approximately 2 pc diameter disk at a PA of about 50°, cospatial with the 700 µm dust continuum emission and the diameter along the short axis of the 3.8–4.8 µm extended emission. The HCN(3-2) traces high density, $10^{4-5}$ cm$^{-3}$, molecular gas. The CO(6-5) observations[40] are dominated by an extended emission of 6 pc in diameter at a PA of 50° highly offset, $\Delta PA = 60°$, from the measured dust continuum emission measured in the 3.8–4.8 µm AMI/JWST observations and cospatial with the dust continuum emission at 700 µm. Weak molecular CO(6-5) emission is detected in the regions of lack of emission ('Holes') in the 3.8–4.8 µm wavelength range.

From the gas tracer observations, we conclude that the 3.8–4.8 µm dust continuum emission in the 'North arc' is spatially coincidental with the [CI] $^3P_1$-$^0P_1$ and CO(6-5) molecular outflows and the H36α ionized outflow. The best spatial coincidence is with [CI] $^3P_1$-$^0P_1$ tracing diffuse atomic gas with a critical density of $n_{cr} = 3.7 \times 10^2$ cm$^{-3}$ [25]. The 3.8–4.8 µm 'North arc' dust continuum emission may be a dusty phase entrained in the outflowing material at 3 pc from the AGN.

We find no spatial correspondence of the $2 \times 5$ pc² extended component at a PA of −70° with any of the molecular or ionized gas tracers. This result may indicate that the 3.8–4.8 µm extended emission is mainly arising from directly radiated dust along the funnel of the obscuring disk and NLR. As mentioned above, the >5 pc dust emission along the east-west direction can be attributed to directly radiated dust by the pc-scale jet[30].

We find that the region with a lack of 3.8–4.8 µm dust emission ('Holes') is spatially coincidental with CO(6-5) molecular emission, the dust continuum emission at 700 µm, the lack of emission in [CI] $^3P_1$-$^0P_1$,

and the 2 pc disk observed at HCN(3-2). The HCN(3-2) is cospatial with the maser disk[31]. The lack of 3.8–4.8 µm dust emission in the northern and southern regions at 8 pc at a PA of 50° from the nucleus may be caused by obscuration effects due to the optically thick obscuring disk around the AGN.

## Discussion

### Origin of the central continuum emission

To quantify the dust temperature and relative contribution of the 3.8–4.8 µm observed morphological structures, we perform a photometric analysis of the central $10 \times 10$ pc² (Methods section 'Photometry'). Using a 2D Gaussian profile to fit the central extended emission, we estimate that $87^{+5}_{-7}$% of the total flux arises from the central $2 \times 5$ pc² at a PA = −70°. We fit a blackbody function that includes a foreground screen of $A_V = 28.5$ mag.[29] to the photometric measurements and estimate a characteristic dust temperature of $532 \pm 10$ K. Using 0.1″ –0.4″ aperture photometry from single-dish telescopes, a characteristic dust temperature of 300 K was measured[28]. This component was attributed to a 2 pc diameter dusty torus obscuring the AGN. The lower dust temperature may be due to the starlight contaminated by the host galaxy. The 8–13 µm extended emission observed by MATISSE obtained dust temperatures of 200–240 K[29]. This component was associated with warm dust in the NLR.

After extracting the central extended emission, we estimate that only 1% of the total emission arises from the 'North arc'. This dust component has a characteristic dust temperature of $1073 \pm 20$ K. Most of the extended emission, $12^{+4}_{-6}$% of the total flux, arises from dust located at > 5 pc from the core, mostly in the east-west direction at a PA of −84° (cospatial with the pc-scale jet) with a characteristic dust temperature of $420 \pm 20$ K. These results indicate that the 'North arc' is an independent dust component not associated with the central extended emission. The > 5 pc extended emission may be dust in the NLR directly radiated by the pc-scale jet and/or AGN–a continuation of the central elongated emission extending to 10s of pc scales[30].

To identify the dominant physical component (i.e., disk vs. wind) of the central elongated emission, we analyze the 1–1000 µm SED by fitting several AGN torus models (Methods section 'Torus models'). The SED was constructed using photometric measurements with a spatial resolution comparable to those by AMI/JWST (Fig. 3; Methods section 'SED'). We use torus-only models with smooth ['Smooth';[41]], clumpy ['CLUMPY';[42,43]], and the 2-phase clumpy torus with inclusion of dust grain sizes ['2phase clumpy'[44]], and the torus+wind model ['CAT3D-WIND'[45]]. For all models, we fixed the inclination of the disk to be edge-on, $i = 90°$ as shown by the maser disk[31], and we include a dust screen as a free parameter.

Including our new AMI/JWST photometric measurements, we find that the torus-only models are statistically preferred to describe the 1–1000 µm SED of the central $2 \times 5$ pc² of Circinus (Fig. 3, Methods section Supplementary 'Table 2'). The largest differences between models arise in the sub-mm wavelength regime (i.e., cold dust component) and the 9.7 µm silicate feature. Note that the sub-mm regime traces the bulk of the dust mass in the torus[46–48], which provides a crucial photometric measurement to characterize the torus models. The 1 − 20 µm SED (including the silicate feature) of the central $2 \times 5$ pc² statistically prefers the '2phase clumpy' model, followed by the smooth torus model. The 1–1000 µm SED of the central $2 \times 5$ pc² statistically prefer the CLUMPY torus models, followed by the CAT3D-WIND model. The CLUMPY torus model underpredicts the 9.7 silicate feature at small scales, except for the large 12.3 pc aperture, indicating that a dust screen may be present along the LOS. This is also found in the MATISSE/VLTI observations[29], showing that a uniform screen of foreground absorption of $A_V = 28.5$ mag. is required to reproduce the silicate feature. The CAT3D-WIND models underpredict the sub-mm dust emission. This is because most of the dust is relocated above and

below the disk (i.e., polar direction), leaving a small and compact dusty disk along the equatorial axis. Most of the SED is then dominated by the hot and warm dust in the base of the outflow. The Smooth models overpredict the sub-mm dust emission, because this model assumes an optically and geometrically thick and monolithic dusty torus. The 2-phase clumpy models overpredict the sub-mm emission. This is because the interclump dust component provides an extra cold emission component to that from the clumps.

To analyze the morphological changes of the continuum thermal emission of the torus, we compute synthetic surface brightness and cloud distribution images using the radiative transfer code Hypercubes of AGN Tori [HYPERCAT;[47,48]]. HYPERCAT uses the CLUMPY torus models with any combination of torus model parameters to generate physically scaled and flux-calibrated 2D images of the dust emission and distribution for a given AGN. We use the statistically preferred CLUMPY torus models model of the 1–1000 $\mu$m central $2 \times 5$ pc$^2$ with an optical depth per cloud of $\tau_V = 20$ with an average of 4 clouds along the radial equatorial plane. We use a distance of 4 Mpc, and a tilt angle on the plane of the sky of 50° cospatial with the orientation along the lack of emission in our AMI/JWST observations. The HYPERCAT images (Methods section 'HyperCAT') are then smoothed and pixelated to match the 3.8–4.8 $\mu$m AMI/JWST, 12.0 $\mu$m MATISSE/VLTI, and 700 $\mu$m ALMA observations. In Fig. 3, we show that the observed morphological changes in the distribution of dust emission from the IR to the sub-mm wavelengths can be reproduced by the CLUMPY torus model with a size of $5 \times 3$ pc$^2$ and a median dust mass of log($M_{dust}[M_\odot]$) = 3.61. (Methods section 'Torus models').

The central $2 \times 5$ pc$^2$ extended component in the 3.8–10 $\mu$m wavelength range can be attributed to thermal emission from directly radiated dust along the funnel of the torus. The lack of 3.8–10 $\mu$m emission perpendicular to the extended component of central $2 \times 5$ pc$^2$ in the 3.8–10 $\mu$m wavelength range can be attributed to optically thick dust along the equatorial axis of the disk. This component is then observed at 700 $\mu$m, which traces the cold dust along the equatorial axis of the disk–co-spatial with the bulk of the dust distribution.

We study the dependency of model fitting with the photometric aperture size (Methods section Supplementary Fig. 6). We find that the 1–1000 $\mu$m SED statistically prefers the CLUMPY torus models for apertures encompassing the central 3 pc extended emission. Using a large aperture of 12.3 pc that contains the 'North Arc' and the emission along the NLR, the smooth torus models best describe the 1–1000 $\mu$m SED, with the clumpy torus models underestimating the NIR and sub-mm fluxes, and the 2-phase clumpy model overestimating the sub-mm fluxes. The 9.7 $\mu$m feature is well described by all models using large-aperture SED. These results imply that there is a diffuse extended emission at >4 pc-scales, not associated with the central torus, that needs to be accounted for. The NIR excess may arise from the outflow and extended emission in the NLR (Fig. 3). This component may be accounted for by the CAT3D-WIND, which is the next preferred model for the 1–1000 $\mu$m SED (Methods section Supplementary Table 2).

In conclusion, the 3–5 $\mu$m emission excess mainly (87%) arises from dust within the 5 pc around the AGN associated with the funnel of the postulated torus. This central extended emission has no spatial correspondence with any of the molecular and ionized outflowing gas at a similar, or better, spatial resolution. The morphological changes of the dust emission are consistent with those expected by a CLUMPY torus with the dust distribution located on a disk of $5 \times 3$ pc$^2$ at a PA of 50°. Thus, most of the mass reservoir is located in the form of an accreting dusty disk that feeds the SMBH of Circinus.

The anisotropic emission measured along the east-west direction, dominated by direct radiation from the pc-scale jet, requires an additional component not associated with the torus[30,49]. This component accounts for 12% of the 3.8–4.8 $\mu$m emission at >5 pc scales, which we attribute to dust in the NLR directly irradiated by the AGN and by the pc-scale jet. Studies[50] using MIRI/MRS/JWST observations of 6 nearby

AGN have shown that the 39–450 pc MIR extended emission accounts for < 40% of the total 10 $\mu$m emission. This MIR emission is attributed to dust located in molecular clouds and/or shocks in the NLR heated by outflows and/or the central AGN.

Our observations show that <1% of the 3.8–4.8 $\mu$m emission arises from dust entrained in the multi-phase outflow labeled as 'North Arc'. This structure has been spatially correlated with molecular and ionized gas at similar, or better, spatial resolution. The 'North arc' may be dust entrained in the outflow, which requires an additional physical component to that of the dusty torus or dust in the NLR. Indeed, an outflowing material has been modeled to explain the anisotropic radiation[30,49].

We show that the combination of the 3–10 $\mu$m and 700–1200 $\mu$m imaging observations is critical to disentangle the dust emission components of the central 10 pc around AGN. In addition, complementary molecular and ionized gas is required to spatially correlate dust entrained in outflows. Our JWST interferometric observations open a new window to disentangle the accreting and outflowing dusty components in the central 10 pc of nearby AGN without the effect of the JWST's PSF, saturation in the imaging mode, and large-scale starlight component of the host galaxy.

## Methods
### Observations
Circinus was observed (ID: 4611; PI: Lopez-Rodriguez, E.) on 20240715 and 20250427 using the Aperture Masking Interferometry [AMI;[32]] mode of the Near Infrared Imager and Slitless Spectrograph [NIRISS;[33]] instrument on the JWST. We performed observations of Circinus and a standard star, HD119164, using the F380M ($\lambda_c$ = 3.827 $\mu$m, $\Delta\lambda$ = 0.21 $\mu$m), F430M ($\lambda_c$ = 4.326 $\mu$m, $\Delta\lambda$ = 0.20 $\mu$m), and F480M ($\lambda_c$ = 4.817 $\mu$m, $\Delta\lambda$ = 0.30 $\mu$m) filters. For all observations, the AMI pupil mask (Supplementary Fig. 1), the SUB80 (80 × 80 px$^2$) array, and the NISRAPID readout pattern were used with a pixel scale of 65 mas px$^{-1}$ and a readout of 75.44 ms. To avoid signal limit (26,000 e$^-$) of the standard star during the observations, we used a setup with 145, 338, and 190 integrations and 4, 4, and 5 groups in the filter sequence of F480M, F380M, and F430M, respectively. This sequence is used to optimize the direction of rotation of the filter wheel and prolong the life of the mechanism. For both Circinus and standard star, the total/effective exposure times are 102/76 s, 57/72 s, and 44/33 s in the F380M, F430M, and F480M filters, respectively. The Stare mode without a dither pattern was used. These are the first two observations of a set of three from this JWST program to rotate the uv-plane of the observations. The V3 position angles are 4° and 87° for the first and second epoch of observations. The uv-plane has rotated by 87°–4° = 83° (Supplementary Fig. 1), which agrees with the 90° with a 10° margin offset between both epochs requested for these observations.

### Data Reduction
We processed the NIRISS AMI observations using the JWST Calibration pipeline (version 1.15.1; CRDS version 11.17.26) and the CRDS context JWST_1258.PMAP. We followed the standard NIRISS AMI data reduction recipe for stages 1 to 4. Stage 1 (CALWEBB_DETECTOR1) produces corrected count rate images ('rate' and 'rateints' files) after performing several detector-level corrections. Stage 2 (CALWEBB_IMAGE2) produces calibrated exposures ('cal' and 'calints' files), where we skip the photometric calibration (photo = False) and resampling (resample = False) of images. These steps produce the interferogram image in units of digital counts shown in Supplementary Fig. 1. Stage 3 (AMI_ANALYZE) is a specific pipeline step for the AMI observations. This step produces the interferometric observables ('ami-oi' files) after computing fringe parameters for each exposure producing an average fringe result of the full observations. The uv coverages of the observations for both epochs are shown in Supplementary Fig. 1. Stage 4 (AMI_NORMALIZE) produces ('amimorn-oi' files) the final normalized interferometric observables

after correcting the science target using the reference standard star. The normalized and calibrated visibilities, $V^2$, and closure phases for both epochs are shown in Supplementary Fig. 2.

Additionally, we obtained calibrated interferometric observables using SAMPip[51]. This software uses a fringe-fitting routine to look for the amplitude and phase solutions that recover the structure of the interferogram, considering the non-redundant mask geometry of NIRISS/JWST. Each frame in the data cubes is fitted individually, and the final squared visibilities and closure phase values are averaged per data cube with their corresponding standard deviations. The observables from the science data cubes are corrected by the instrumental transfer function using the standard star HD 119164. The calibrated observables are stored in standard OIFITS files[52] for posterior analysis.

## Image reconstruction

We reconstructed the Circinus images at each filter using SQUEEZE (GitHub repository of SQUEEZE: https://github.com/fabienbaron/squeeze) [version 2.7;[36]]. This algorithm has successfully been used to reconstruct the NIRISS AMI observations of the Wolf-Rayet, WR 137[53]. SQUEEZE reconstruction image algorithm uses a Markov Chain Monte-Carlo (MCMC) approach to explore the imaging probability space using the interferometric observables with its associated uncertainties. Using the SAMPip outputs, SQUEEZE images were recovered using a pixel grid of $129 \times 129$ px$^2$ (FOV = $1.29 \times 1.29$ arcsec$^2$), with a pixel scale of 10 mas. For the reconstruction, we used two regularization functions, a Laplacian and the L0-norm, with the following hyperparameter values $\mu_{La} = 500$ and $\mu_{L0} = 0.2$, respectively. With these parameters, the reconstructions converged with $\chi^2_\nu$ close to unity. To characterize the signal-to-noise ratio (SNR) of the images, we recovered 100 images per data cube with different samples of the observed uv-plane. For this procedure, we randomly sampled the uv frequencies of the interferometer by changing their weights; at the same time, we kept the total number of uv points constant. Each image is the result of the MCMC from SQUEEZE from a random sample of the uv plane with a specific uv weight. Finally, we averaged the 100 different reconstructions per wavelength to construct the final image. We computed the dirty beam of each filter shown in Supplementary Fig. 3. The angular resolutions in the reconstructed images of Circinus are $93 \times 88$ mas$^2$ ($1.9 \times 1.8$ pc$^2$), $105 \times 101$ mas$^2$ ($2.1 \times 2.0$ pc$^2$), and $123 \times 116$ mas$^2$ ($2.5 \times 2.3$ pc$^2$) in the F380M, F430M, and F480M filters, respectively, which corresponds to the theoretical angular resolution of $\lambda/2B$ by the interferometric observations.

To estimate the validity of the reconstructed images compared with the calibrated observables, we computed the synthetic interferometric observables from each one of the recovered images per wavelength. The mean value and the standard deviation of the synthetic observables are shown in Supplementary Fig. 4. It can be observed that all data points from the images are consistent with the data within $1\sigma$. Similarly, to estimate the statistically significant features with SNR above the noise level in the images, we estimated their noise statistics ($\mu_{noise}$, $\sigma_{noise}$) using all the pixels values outside a box of $40 \times 40$ px$^2$ ($8 \times 8$ pc$^2$) centered in the image. The interferometric observables of those filtered images are consistent within $1\sigma$ of the reported synthetic observables, allowing us to trust the significance of the recovered morphology.

## WCS correction

The reconstructed images do not have a world coordinate system (WCS) associated with them. However, the interferogram pattern (Supplementary Fig. 1) has the WCS from the JWST observations. Thus, we use the sky coordinates of the peak from the interferogram pattern as the sky coordinates of the peak pixel from the reconstructed images at each filter. Here, we assume that the peak pixel of the reconstructed image is the position of the AGN, which dominates the IR emission of the object in both the interferogram pattern and the reconstructed

images. A small WCS shift was then performed to align the AMI/JWST observations with the peak emission of the 1200 $\mu$m ALMA observations (Fig. 2). The 1200 $\mu$m ALMA observations trace the radio synchrotron emission from the jet and AGN, we assume this is the 'true' center of the AGN in our work.

## Flux calibration

The standard star HD119164 ($F_{12\,\mu m} = 1.2$ Jy) was observed immediately after the science object using the same configuration as that for the Circinus galaxy. We took observations of the same standard star as previously used by the interferometric observations of Circinus with MATISSE/VLTI at L, M, and N-bands[27,29]. The standard star serves to perform the flux calibration and final visibilities of the science object. The flux calibration of the final reconstructed image of the science object was computed as:

$$F_{obj}^{cal}(\lambda)\,[Jy] = F_{obj}^{norm}(\lambda) \times \frac{F_\star(\lambda)\,[Jy]}{F_\star^T(u=0, v=0, \lambda)\,[ADU]} \times F_{obj}^T(u=0, v=0, \lambda)\,[ADU],$$

$$(1)$$

where $F_{obj}^{norm}(\lambda)$ is the normalized reconstructed image of the science object with a total flux equal to unity, $F_\star(\lambda)$ is the total flux of the standard star in units of Jy, $F_\star^T(u=0, v=0, \lambda)$ is the total flux of the zero-baseline of the standard star in units of counts (i.e., ADU: analog digital unit), and $F_{obj}^T(u=0, v=0, \lambda)$ is the total flux of the zero-baseline of the science object in units of counts (i.e., ADU). All these fluxes are at a given wavelength, $\lambda$.

$F_\star(\lambda)$ was estimated using the spectral type, G8II, of the standard star, scaled to have a flux of 1.2 Jy at 12 $\mu$m[27,29]. Then, we estimated the total flux of the standard star within the bandpass (The NIRISS throughputs can be found at https://jwst-docs.stsci.edu/jwst-near-infrared-imager-and-slitless-spectrograph/niriss-instrumentation/niriss-filters#NIRISSFilters-NIRISSsystemthroughput) of the NIRISS/AMI filters to be 8.55, 7.09, and 5.95 Jy at F380M, F430M, and F480M, respectively. $F_\star^T(u=0, v=0, \lambda)$ was estimated using the total flux of zero-baseline from the image of the mirrored Hermitian counterparts in this uv-plane coverage, or Modulation Transfer Function [MTF; see Fig. 1 by ref. 32]. The zero-baseline contains the total flux of the observations. We computed the total flux from the central peak of the MTF image using two methods. First, we perform aperture photometry with a radius of 3.5 pixels. Second, we fit a 2D Gaussian profile with two free parameters: the amplitude and the FWHM, which is assumed to be axisymmetric. We estimate that the aperture photometry (i.e., first method) misses about 12–22% of the flux arising from the wings of the 2D Gaussian profile. We use the total flux of the zero-baseline estimated with the 2D Gaussian fitting profile. NIRISS AMI mode has a photometric calibration uncertainty (NIRISS AMI photometric calibration: https://jwst-docs.stsci.edu/depreciated-jdox-articles/jwst-data-calibration-considerations/jwst-calibration-uncertainties#JWSTCalibrationUncertainties-Photometriccalibration.10) of 5% in the F380M and F430M filters and 8% in the F480M filter.

## Emission line contribution

To estimate the potential contribution of spectral features within the filters, we use synthetic photometry on both the observed spectra and the feature-free continuum spectra of local AGN. First, we establish a baseline representing the continuum emission from the central spectrum by fitting feature-free continuum anchor points with straight lines [e.g.,[54]]. Using the fitted baseline, we then perform synthetic photometry for the NIRISS imaging bands by convolving the spectra with the corresponding filter transmission curves (http://svo2.cab.inta-csic.es/svo/theory/fps/index.php?mode=browse&gname=JWST&gname2=NIRISS&asttype=, as in ref. 55 (see also Donnelly et al.[56]). The main features contributing to the F380M, F430M, and F480M filters include gas-phase and icy molecular bands such as the

$^{12}$CO (4.45 − 4.95 μm) molecular gas-phase absorption band and the 4.27 μm stretching mode of the $CO_2$ ice [e.g.,[54,57,58]]. We utilize local AGN (NGC 3256 and NGC 7469) observed with MRS/JWST from the Director's Discretionary Early Release Science Program #1328 (PIs: L. Armus & A. Evans) to calculate the fractional contribution of the continuum to the photometry of type 1 and type 2 AGN. For type 2 AGN, we find a continuum contribution of 94%, 71%, and 84% in the F380M, F430M, and F480M filters, respectively. Note that these continuum contributions should be considered a lower limit, as the lines are known to be stronger in luminous IR galaxies, such as NGC 3256, which was used in this estimation. In the case of the type 1 NGC 7469, the continuum dominated the emission in all the filters used in this work.

## Archival observations

For our imaging analysis, we use the following archival observations. NACO/VLT images at L'-band (3.8 μm, $\Delta\lambda = 0.62$ μm) with an FWHM of 0.12″[28]. VISIR/VLT at N-band (10.5 μm; $\Delta\lambda = 0.01$ μm) with an FWHM of 0.3″[30]. MATISSE/VLTI at 12.0 μm with an FWHM of 10 mas[29]. Dust continuum emission at 700 μm and non-thermal emission at 1200 μm using ALMA with beam sizes of $107 \times 64$ mas$^2$ at a PA = 74° and $27 \times 24$ mas$^2$ and PA = 15°, respectively[25]. Dust and radio continuum emission at 890 μm with a beam size of $100 \times 80$ mas$^2$ and PA = −1.7° (ALMA ID 2022.1.00222.S), [CI] $^3P_1$-$^0P_1$ with a beam size of $119 \times 76$ mas$^2$ and PA = 75°, H36α with a beam size of $62 \times 43$ mas$^2$ and PA = −8°, and HCN(3-2) with a beam size of $29 \times 24$ mas° and PA = 20°[25]. CO(6-5) with a beam size of $95 \times 66$ mas$^2$ and PA = 34°[40].

## Photometry

We perform photometric measurements using a circular aperture and an elongated 2D Gaussian. We compute circular aperture photometry with (a) a diameter equal to the FWHM at each wavelength, (b) a fixed aperture equal to the lowest resolution of our observations, i.e., 123 mas (2.5 pc at 4.8 μm), and c) a fixed aperture encompassing the full extended emission of the Circinus, e.g., 640 mas (12.3 pc). In addition, to optimize the extraction of the fluxes from the elongated emission, we performed photometric measurements using a 2D Gaussian profile. The 2D Gaussian profiles are fixed at the location of the peak pixel at each wavelength and have four free parameters: the x and y axes of the FWHM, the PA, and the total amplitude of the peak of the 2D Gaussian profile. We computed a Markov Chain Monte Carlo (MCMC) approach using the No-U-Turn Sampler [NUTS;[59]] method in the PYTHON code PYMC3[60]. We set flat prior distributions within the ranges of $x = y = [0, 3] \times$ FWHM at a given wavelength, $\theta = [0, 180]°$ East of North, and $I_0 = [0, 1]$ (peak has been normalized to unity). We run the code using 5 chains with 5,000 steps and a 1000 burn-in per chain, which provides 20,000 steps for the full MCMC code useful for data analysis. Supplementary Fig. 5 shows the best fits of the 2D Gaussian models per wavelength and the residuals. Supplementary Table 1 shows the photometric measurements of all the methods described above.

## SED

We took the 1−20 μm SED with apertures of 0.1−0.4″ used by Stalevski et al.[30], and added the 3.8−4.8 μm AMI/JWST photometric points from our analysis (Supplementary Table 1), and the photometric measurements of the central 123 mas using the 700−1200 μm ALMA observations shown in Fig. 2. The 1−1000 μm SED is shown in Fig. 3, and in Supplementary Figs. 6, and 7. To estimate the relative contribution of non-thermal synchrotron emission at 700 μm, we use the radio observations from 3 to 20 cm observed by the Australia Telescope Compact Array (ATCA)[61]. We estimate that the non-thermal synchrotron emission at 700 μm contributes <50%. Note that the ATCA data have a low angular resolution 20″, compared to the 100 mas resolution from the ALMA observations. Thus, this relative contribution is an overestimated upper-limit to the synchrotron emission at 700 μm.

## Torus models

We took four torus models comprising several geometries with the main goal of distinguishing between a disk-like or wind-like structure[20,62]. For all models, we fixed the inclination of the disk to be edge-on, $i = 90°$, and let the dust screen be a free parameter, E(B-V).

Smooth[41] torus model uses a torus-like geometry with a smooth dust distribution. The torus parameters are: $i$ is the viewing angle toward the torus, $\sigma$ is the half opening angle of the torus, $\gamma$ and $\beta$ are the exponents of the logarithmic azimuthal and radial density distributions, respectively, $Y = R_o/R_i$ is the ratio between the outer and inner radii of the torus, and $\tau_V$ is the edge-on optical depth at 0.55 μm.

CLUMPY[42,43] torus model uses a clumpy distribution distributed in a torus-like structure. The free parameters are: $i$ is the viewing angle toward the torus, $N_0$ is the mean number of clouds radially across the equatorial plane, $\sigma$ is the half opening angle of the torus width measured from the equatorial plane, $Y = R_o/R_i$ is the ratio between the outer and inner radii of the torus, $q$ is the slope of the radial density distribution, and $\tau_V$ is the optical depth at 0.55 μm of individual clouds.

2-phase clumpy[44] torus model uses a torus geometry with high-density clumps and low-density, and a smooth interclump dust component. The free parameters are: $i$ is the viewing angle toward the torus, $\sigma$ is the half opening angle of the torus, $p$ and $q$ are the indices that set the dust density distributions along the radial and polar directions, respectively, $Y = R_o/R_i$ is the ratio between the outer and inner radii of the torus, $\tau_V$ is the average edge-on optical depth at 0.55 μm, and maximum dust grain sizes, $P_{size, max}$. The extinction by dust grains was taken from ref. 63 and website https://heasarc.gsfc.nasa.gov/xanadu/xspec/manual/node291.html.

CAT3D-WIND[45] torus model uses a clumpy disk and a dusty polar outflow. The free parameters are: $i$ is the viewing angle toward the torus, $N_0$ is the number of clouds along the equatorial plane, $a$ is the exponent of the radial distribution of clouds in the disk, $\theta$ is the half-opening angle of the dusty wind, $\sigma_\theta$ is the angular width of the hollow dusty wind cone, $a_w$ is the index of the dust cloud distribution power-law along the dusty wind, $h$ is the height of the inner edge of the torus, and $fwd$ is the wind-to-disk ratio.

We fit each of the torus models to the 1−1000 μm SED with and without the AMI/JWST photometric measurements (Supplementary Fig. 7). We use the same fitting routine described in González-Martín et al.[62], which uses a $\chi^2$ minimization approach. The model parameters and 1$\sigma$ uncertainties associated with the best-fit model are shown in supplementary Table 2. We estimate the $\chi^2$ of the models within the 1−1000 μm SED, $\chi^2_{ALL}$ and within the 1−20 μm SED, $\chi^2_{IR}$.

We estimate the dust mass of the two favored torus models: CLUMPY and 2-phase torus using the best-fit parameters of the fixed aperture (123 mas; 2.5 pc). The median dust masses are $\log(M_{dust}[M_\odot]) = 3.61$ and $\log(M_{dust}[M_\odot]) = 3.74$ for the CLUMPY and 2phase models, respectively. For the CLUMPY torus modes, $\sigma$ and $Y$ are upper limits, so the maximum torus mass is $\log(M_{dust}[M_\odot]) = 4.61$. For the 2-phase torus model, we have a lower-limit for $p > 1.3$ and an upper limit for $q < 1.5$. Using both limits, we estimate a mass range of $\log(M_{dust}[M_\odot]) = 3.38 - 5.54$. Note that in all cases $q$ is at the upper-bound of its range, which indicates that the torus has its mass mainly concentrated on the inner side of the torus (i.e., radial distribution of clouds: $r^{-q}$, where $r$ is the radial distance). For comparison, the molecular masses of the torus are estimated to have a median of $\log(M_{dust}[M_\odot]) = 5.77$[64]. This mass is estimated using CO observations using ALMA of nearby AGN with higher bolometric luminosities $\log(L_{bol}[ergs^{-1}]) = 44.0$ than the Circinus galaxy $\log(L_{bol}[ergs^{-1}]) = 43.6$. Lower dust mass is expected in the torus of Circinus.

## HyperCAT

The emergent thermal emission as a function of wavelength for the best fit of the CLUMPY torus model at the native resolution (1.2 mas) of the model is shown in Supplementary Fig. 8. Note that the thermal

emission distribution changes as a function of wavelength. At $4.3\,\mu m$ and $10\,\mu m$ the emission is along the funnel of the torus, while the $700\,\mu m$ thermal emission is along the equatorial axis of the torus. At all wavelengths, the emission drops at the core due to the column density along the LOS, and the emission peaks at the inner walls of the torus, directly radiated by the AGN and directly viewed by the observer LOS.

## Data availability

JWST data is available in MAST. The JWST data used in this study are available in the MAST database under accession code 4611 [https://mast.stsci.edu/portal/Mashup/Clients/Mast/Portal.html]. The reconstructed image using our image reconstruction will be available upon request. Source data are provided with this paper.

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

## Acknowledgements

This work is based on observations made with the NASA/ESA/CSA James Webb Space Telescope. The data were obtained from the Mikulski Archive for Space Telescopes at the Space Telescope Science Institute, which is operated by the Association of Universities for Research in Astronomy, Inc., under NASA contract NAS 5-03127 for JWST. These observations are associated with program #4611. Support for program #4611 was provided by NASA through a grant from the Space Telescope Science Institute, which is operated by the Association of Universities for Research in Astronomy, Inc., under NASA contract NAS 5-03127. E.L.-R. thanks support by the NASA Astrophysics Decadal Survey Precursor Science (ADSPS) Program (NNH22ZDA001N-ADSPS) with ID 22-ADSPS22-0009 and agreement number 80NSSC23K1585. J.S.-B. acknowledges the support received by the UNAM DGAPA-PAPIIT project AG 101025.

## Author contributions

E.L.R. led the JWST proposal, data reduction, image reconstruction, scientific analysis, and manuscript writing and editing the full manuscript. J.S.-B performed the image reconstruction, interferometric analysis, and wrote the image reconstruction section. O.G.-M performed the torus models fitting and scientific interpretation. R.N. supported the JWST proposal, the HyperCAT images, and scientific interpretation. R.M.L. supported the JWST proposal, JWST observations, and image reconstruction. D.T. supported the JWST proposal, JWST observations, and AMI data reduction. I.G.M. supported the emission line contamination analysis and scientific interpretation. J.H.G. supported the JWST proposal and JWST observations. M.J.H. supported the JWST proposal.

## Competing interests

All authors declare no competing interests.
