## [Transparent Peer Review file · Nature Communications]

JWST interferometric imaging reveals the dusty torus obscuring the supermassive black hole of the Circinus galaxy

Corresponding Author: Dr Enrique Lopez Rodriguez

Version 0:

Reviewer comments:

Reviewer #1

(Remarks to the Author)

In this paper, the authors are investigating the physical properties of the central region of the AGN by high-resolution observations with JWST in comparison with other wavelengths' observations.

A unique point in this work is to utilize an aperture-masking interferometric mode, which enables the authors to conduct highest resolution observations of this type of objects.

Although I have no serious concerns about the observational results themselves, I have significant skepticism regarding their interpretation.

One of the main claims is that the dominant component in the 3-5 μ m wavelength range is the dust distributed in the funnel regions of the torus. On the other hand, the torus, which is regarded to have a larger dust mass, is not directly observed; the authors speculate that this is because the dust associated with the torus is optically thick (Lines 400-403). Regarding this point, I have the following questions.

(1)

The authors implicitly assume that a sizable fraction of the dust grains comprising the funnel are distributed in front of the torus, because the emission from the center is not obscured.

If the dust grains are supplied from the torus by ejections or disk winds, how are they in front of the torus.

(2)

The hypothesized torus is not observed even with the 1.2 mm wavelength by ALMA. Is it possible to put constraints on the properties of the dust (total mass, size, and temperature) in the torus?

Reviewer #2

(Remarks to the Author)

-Summary

The authors present high resolution observation of the nuclear region of Circinus using a novel aperture-masking interferometric mode onboard the JWST. The work is really interesting because present a novel technique that improves the spatial resolution and enhances the understanding of the nature of the torus. The authors compare different bands, both with lower and higher resolution than AMI, and then they perform the photometry, decoupling the unresolved contribution and the resolved contribution from the disk and an emission arising from an arc-like structure. They build SED models and study which one better describes the observations as a function of the aperture size.

The work is original, few works resolve the dusty torus of AGN, mostly with this novel technique. The manuscript is well-structured, the analysis is clear and well described even if in some passages could be provided more details and discussion. I would recommend publication of the paper in Nature Communication.

General major comments:

-Photometry and SED: Do the black dots in the SED figures (Fig.3,10) correspond to same apertures of AMI or you studied the dependence of the aperture just in function of the AMI photometric points. In the second case, can the best model change? (Line 602,603)

General minor comments:

-South Arc: The author refers to the North Arc as a features detected at $> 16\sigma$ but is not clear if the other features are real features or not, because there might be also a South Arc that could make the morphology also more interesting. At least should be discussed a bit more than Lines 162-164.
-In the section "Multi Phase Components Of The Extended Emission", the author compares the multiphase emissions, but it should be highlighted more the difference between this emission, what they trace and discuss a bit more. For example Lines 237-238, 257-258, 403-404 (see further comments).
-Figures: Since the author compare different observations, when they note a cospatial structure between two bands, could be better to add contours.
-Check tenses.

Here I have some specific comments: some are typo that I report for improving the quality of the manuscript, others are clarifications that could be added.

Line 20: specify the "possible resolution"

Line 21-24: specify which emission

Line 47: specify properties such as the temperature of the warm, optically thin dust in the MIR emission.

Line 78: define position angle.

Line 84: the the typo

Line 87: PA

Lines 80-96 is not clear if the extended emission is or is not observed in MATISSE observations. Make the point more clear.

Line 144: cite some of the brightest line.

Lines 167-168 cite NIRCam/JWST papers.

Line 172: Fig.

Line 190: Fig. 2 (e)?

Lines 197-202: Why are MATISSE data in your band not presented? I think it will benefit to understand the differences between AMI and MATISSE.

Lines 201, 212: Clarify the definition of PA and how is computed.

Line 216: Is this due to some physical processes?

Lines 222-228: From the grey contours, also in the eastern region seems that the synchrotron emission shows an arc feature and it seems cospatial just in the first parsec.

Lines 228-230: Consider overlaying MATISSE N-band contours in Fig. 2h.

Lines 235-237: Is this due to some physical processes?

Lines 257-258: Why the extended emission is just toward the base of the North Arc and there is a lack of emission in the eastern region.

Lines 263: specify what the HCN(3-2) emission trace. Could be worth to note the lack of emission in the centre ?

Lines 291-295: Why not citing also [CII]?

Line 308: In line 154 the PA is -64deg. Clarify how you compute this.

Line 310: How this dust temperature compares with the temperature reported in Lines 193-195?

Line 334: Which observations you are using for your photometric SED.

Lines 369-375: Rephrase avoiding repetition.

Lines 403-404: Explain also why it is not seen at 890 μm

Line 411: The used aperture is 8pc or 12.3pc as indicate in App.9 (Fig10)?

Line 412: 'North Arc'

Lines 420-421: Truncated phrase (probably refers to Fig. 3?)

Lines 441-442: References not linked.

Lines 558: Say why you align with the 1200 μm ALMA.

Lines 573: fig. -> Fig.

Lines 576: "Winds"?

Lines 587-592: Say which instrument you used to compute the contribution of the lines.

Line 603: Specify which has lower resolution since you used many observations.

Fig.8: Optimize space in the figure using just one colorbar, put logarithm in the label and use arcsec instead of pixel as in other Figs.

Reviewer #3

(Remarks to the Author)

I believe this work represents an important step forward in the study of active galactic nuclei and their environments. I recommend publication following a few relatively minor changes and/or clarifications.

#####

Major Comments

#####

In the discussion about directly illuminated PA~64 deg dust in the NLR, the authors cite Stalevski et al. (2017) in the context of radio jet illumination. However in that work (and Stalevski et al. 2019), the illumination is instead attributed to the warped accretion disk. If the authors prefer the jet-based interpretation, it would be helpful to expand the discussion and clearly justify this deviation from earlier interpretations.. Moreover, it may be helpful to note that the maser disk is thought to be warped (Greenhill 2003), causing anisotropic illumination of the dusty structures. This is explored in Jud et al. (2016) and used as the reasoning for the preferential illumination of one edge of the outflow cone in Stalevski et al. (2017, 2019).

In fitting the SED with RT models, the authors assume the disk to be perfectly edge-on. Previous authors, such as Stalevski et al. (2019) required a slight inclination (~85 deg) in order to explain bright MIR emission (rather than absorption) along the torus PA. Do the authors expect the results of the SED fitting (and which sets of models work best) change when using the previous value of ~85 deg instead of 90 deg?

The authors claim that the torus-only models are statistically preferred to describe the 1-1000 micron SED. Table 2, however, includes many chi2 values and in many cases the torus-only models are not preferred (ie, exhibiting the smallest chi2). The authors should clarify which dataset (of the four) they use to make this claim – i.e., without JWST or with JWST and which aperture size.

The choice to fit a pure blackbody rather than a graybody or a blackbody with foreground absorption is not fully justified. It may be helpful to clarify the wavelength range used for this fit, and to consider whether including a foreground extinction screen (e.g., $A_v \sim 28$ mag) would materially affect the inferred temperature. Additionally, typical ISM extinction will change the M-band fluxes quite a bit relative to the L-band (e.g., Fig 3; Schartmann et al. 2005) even relatively far from the silicate feature at ~10 micron. This changing MIR color will possibly impact a fitted temperature result (only quantitatively, and the qualitative results of the paper will remain).

#####

Minor Comments

#####

It is unclear where the uncertainty (and therefore significance of features) on the final images comes from. The SQUEEZE results are from an MCMC, meaning that a large number of images are created for each bootstrap sampling of the uv weights. The authors mention recovering 100 different images and averaging those, but are each of those 100 images already an average of the SQUEEZE MCMC sampling?

The authors mention that the North Arc is filtered out in the MATISSE images. Rather than listing the smallest scales that the MATISSE observations could attain, it might be useful to list the largest scales the MATISSE observations were sensitive to. This may aid in interpretation of the apparent M-band PA discrepancies between the two datasets.

Could the authors please explain why they are displaying/comparing to the 10.5 micron image from Isbell et al. (2022)? The 10.5 micron image is still within the rather deep, broad silicate feature they report, and so it is not a continuum image. The 11.3 or 12.0 micron image would better represent the continuum. If this image was selected to match the 10 micron image of e.g., Stalevski et al. (2017), the authors could instead also use the 11.9 um image from that paper.

The flux densities are apparently negative in Fig 8. Are they meant to represent fractions of the peak in logarithmic spacing?

Line 421 appears to reference Figure 6, but based on the context, this may have been intended to refer to Figure 3 or 10.

It is unclear to me at which position the fluxes/temperatures of the northern arc are being measured. Adding apertures to Fig 8 would clarify the location of the temperature estimate (~720 K) and the scale of the structures being measured.

The authors discuss the discrepancy between most of the models and the SED at ~5um (shown in e.g. Fig 10) and state that CAT3D-WIND can help describe this. Fig. 10 shows that this discrepancy becomes less significant with larger aperture sizes. Does this indicate that the feature is arising from smaller scales?

It may be worth pointing out more explicitly that while the chi2 comparisons to the various models preferred the Torus alone, the Hypercat comparisons and observed morphology point almost directly to a disk+wind scenario. In the works of Gonzalez-Martin et al. (2019) and Isbell et al. (2021), the Nenkova2008 models are shown to cover a large parameter space in NIR/MIR colors – larger than the other torus(+wind) models analyzed. Could this play a role in the statistical preference for torus-only models via chi2 in this paper, when the morphology and Hypercat comparisons indicate otherwise?

Version 1:

Reviewer comments:

Reviewer #1

(Remarks to the Author)

I appreciate the authors for thoroughly revising the manuscript.

I found that the manuscript is almost satisfactory but I have a remaining question regarding the point 2 in my original draft.

>(2) The hypothesized torus is not observed even with the 1.2 mm wavelength by ALMA. Is it possible to put constraints on the properties of the dust (total mass, size, and temperature) in the torus?

>The dusty torus is observed with ALMA at 700 μm (Figure 2-f), and the molecular torus is observed with ALMA at CO(6-5) (Figure 2-j) and HCN(3-2) (Figure 2-l). Our HyperCAT models at 700 μm reproduce the thermal emission of the dusty torus as shown in Figure 3. Note that the 1.2 mm AMA observations are dominated by the radio synchrotron emission from the radio jet, which is shown in Figure 2-h and the SED in Figure 9.

I cannot find an answer to my question about the constraints on dust properties.

Is it impossible to put a constraint on the dust mass and size in the torus?

Reviewer #2

(Remarks to the Author)

The author addressed all the comments.

Few minor comments:

20: ,, and units.

295 trace -> traces

Reviewer #3

(Remarks to the Author)

The authors have satisfactorily answered my major concerns. In particular, the inclusion of the apertures in Fig. 8 and the description of the image reconstruction enhance clarity. I thank the authors for their detailed responses.

The authors are correct in pointing out that after the inclusion of the foreground dust screen, the temperatures are qualitatively the same. Outside the scope of this paper, the much higher inferred North Arc temperature could have interesting implications for dust density/clumpiness in the so-called polar wind.

Regarding my previous comments about the torus-only vs disk+wind model as inferred from the χ^2 comparisons and to HyperCAT: I did not fully appreciate the constraints that the ALMA torus provided to the interpretation. With the revised text and the authors' responses, it is now clearer to me how the SED in combination with the morphological constraints led to the authors' interpretations. In particular, the addition of Figure 11 helps illustrate the regimes in which the "torus" dominates or in which the "wind" dominates.

Version 2:

Reviewer comments:

Reviewer #1

(Remarks to the Author)

I appreciate the authors for the further revision to the manuscript. I have found that the revised draft is satisfactory to me. I recommend the paper for publication.

Dear Anonymous Referees,

Thank you for the detailed reading and comments on this work. We have implemented all your suggested changes and answered your comments on a point-by-point basis below. The new version has the implemented revisions in bold.

Reviewer #1 (Remarks to the Author):

In this paper, the authors are investigating the physical properties of the central region of the AGN by high-resolution observations with JWST in comparison with other wavelengths' observations.

A unique point in this work is to utilize an aperture-masking interferometric mode, which enables the authors to conduct highest resolution observations of this type of objects.

Although I have no serious concerns about the observational results themselves, I have significant skepticism regarding their interpretation.

One of the main claims is that the dominant component in the 3-5 μ m wavelength range is the dust distributed in the funnel regions of the torus. On the other hand, the torus, which is regarded to have a larger dust mass, is not directly observed; the authors speculate that this is because the dust associated with the torus is optically thick (Lines 400-403). Regarding this point, I have the following questions.

These conclusions are supported by the analysis performed using the radiative transfer model presented in this section and Figure 3. Specifically, we computed a radiative transfer model of the clumpy torus that best described the SED using HyperCAT (lines 376-396). These models provide the 2D images of the emergent thermal emission as a function of wavelength and the dust mass distribution. These models are displayed in Figure 3 and show how the 4.3 μ m image is dominated by the hot dust along the funnel of the torus, and the 700 μ m image is dominated by cold dust along the equatorial axis of the torus. Note that the dust distribution is along a position angle of 50° East of North.

(1) The authors implicitly assume that a sizable fraction of the dust grains comprising the funnel are distributed in front of the torus, because the emission from the center is not obscured. If the dust grains are supplied from the torus by ejections or disk winds, how are they in front of the torus.

This is not assumed in this work. The interpretation, based on the radiative transfer model, is that the AGN illuminates the funnel of the torus, which is the dominant, and only, thermal emission at 4.3 μ m and 10 μ m. The white contours in Figure 3 show this emission. Note that the synthetic thermal emission (white contours) is the emission after smoothing using a 2D Gaussian profile with a FWHM equal to the AMI and MATISE observations. This may create an illusion that there is dust in front of the torus, but it's just a resolution effect. This effect is clearly seen in Figure 14 by Nikutta et al. 2021a (<https://ui.adsabs.harvard.edu/abs/2021ApJ...919..136N/abstract>). To clarify this potential confusion, the emergent thermal emission and cloud distributions at the original resolution of the model are shown in the Method Section 'HyperCAT'. Note that the thermal emission in all cases drops at the core,

which is indicative of the column density along the LOS. Also note that the torus walls dominate the emission at the very center of the image, which, after smoothing, may appear as if there is also dust in front of the torus. This is not the case; this emission is the dust in the walls of the torus directly viewed by the observer. These comments are now included in the manuscript.

(2) The hypothesized torus is not observed even with the 1.2 mm wavelength by ALMA. Is it possible to put constraints on the properties of the dust (total mass, size, and temperature) in the torus?

The dusty torus is observed with ALMA at 700 μm (Figure 2-f), and the molecular torus is observed with ALMA at CO(6-5) (Figure 2-j) and HCN(3-2) (Figure 2-l). Our HyperCAT models at 700 μm reproduce the thermal emission of the dusty torus as shown in Figure 3. Note that the 1.2 mm ALMA observations are dominated by the radio synchrotron emission from the radio jet, which is shown in Figure 2-h and the SED in Figure 9.

Reviewer # 2 (Remarks to the Author):

-Summary

The authors present high resolution observation of the nuclear region of Circinus using a novel aperture-masking interferometric mode onboard the JWST. The work is really interesting because present a novel technique that improves the spatial resolution and enhances the understanding of the nature of the torus. The authors compare different bands, both with lower and higher resolution than AMI, and then they perform the photometry, decoupling the unresolved contribution and the resolved contribution from the disk and an emission arising from an arc-like structure. They build SED models and study which one better describes the observations as a function of the aperture size.

The work is original, few works resolve the dusty torus of AGN, mostly with this novel technique. The manuscript is well-structured, the analysis is clear and well described even if in some passages could be provided more details and discussion. I would recommend publication of the paper in Nature Communication.

General major comments:

-Photometry and SED: Do the black dots in the SED figures (Fig.3,10) correspond to same apertures of AMI or you studied the dependence of the aperture just in function of the AMI photometric points. In the second case, can the best model change? (Line 602,603)

The black dots are the literature photometric points with equal or better resolutions, 0.1-0.4", than the AMI observations. These photometric points were extracted from Staleski+2017 (<https://ui.adsabs.harvard.edu/abs/2017MNRAS.472.3854S/abstract>). We added the aperture range in the methods section. Note also that the 700-1200 μm photometric points were obtained from the ALMA observations used in this work and using the aperture as described in the Method section 'SED'.

Regarding the second point, Figure 10 shows the changes of the torus models as a function of the SED without AMI data, and with AMI data with several apertures. The results of this study are discussed in lines 344-375.

General minor comments:

-South Arc: The author refers to the North Arc as a features detected at $> 16\sigma$ but is not clear if the other features are real features or not, because there might be also a South Arc that could make the morphology also more interesting. At least should be discussed a bit more than Lines 162-164.

Indeed, these features are at a 4σ level and they are also persistent in the individual visits, combined visits, as well as in the bootstrapping techniques. As these features account for $<1\%$ of the total flux, we decided to briefly mention them in this manuscript.

The JWST took the 3rd visit of this program during the submission of this manuscript and it was further tagged as having astrometry issues. This 3rd visit is at an angle of 45 degrees, which provides another UV plane coverage. Given the high SNR and the large uv-coverage from the 2 visits used in this manuscript, we decided to use the 3rd visit for a further follow-up manuscript. A dedicated study investigating the astrometry issues is required. It is also required to investigate the effect of the uv coverage as a function for the image reconstruction. Having combined the 3 visits, then the extended emission mentioned by the referee is expected to have a >5 sigma significance, which provides a statistically robust detection excellent for detailed analysis of the role of dust on the 10 pc scales. We have included a sentence in lines 162-165 to quantify and acknowledge these features given the tight constraints on text.

-In the section "Multi Phase Components Of The Extended Emission", the author compares the multiphase emissions, but it should be highlighted more the difference between this emission, what they trace and discuss a bit more. For example Lines 237-238, 257-258, 403-404 (see further comments).

This section now includes slightly more details on each of these images given the text constraints.

-Figures: Since the author compare different observations, when they note a cospatial structure between two bands, could be better to add contours.

We appreciate this comment to improve the figures. We tested several options by making specific contours extra thick or adding more labels, but the multi-panel figure became confusing. As the text describes the comparison, we ensured to always explicitly refer to the feature discussed, labeled in Figure 1, and the main PAs labeled in Figure 2.

Some examples are here:

1. Using labels:

2. Using contours

-Check tenses.

The manuscript has been edited.

Here I have some specific comments: some are typo that I report for improving the quality of the manuscript, others are clarifications that could be added.

Line 20: specify the "possible resolution"

DONE

Line 21-24: specify which emission

DONE

Line 47: specify properties such as the temperature of the warm, optically thin dust in the MIR emission.

DONE

Line 78: define position angle.

DONE

Line 84: the the typo

DONE

Line 87: PA

DONE

Lines 80-96 is not clear if the extended emission is or is not observed in MATISSE observations. Make the point more clear.

DONE

Line 144: cite some of the brightest line.

These lines are quoted in the Methods section 'Emission line contribution'.

Lines 167-168 cite NIRCam/JWST papers.

DONE

Line 172: Fig.

DONE

Line 190: Fig. 2 (e)?

DONE

Lines 197-202: Why are MATISSE data in your band not presented? I think it will benefit to understand the differences between AMI and MATISSE.

These datasets are not publicly available and the FOV of the MATISSE in L-band is very small. Any comparison is vastly in different scales, we refer to Isbell+2023 (Fig. 4, <https://ui.adsabs.harvard.edu/abs/2023A%26A...678A.136I/abstract>).

Lines 201, 212: Clarify the definition of PA and how is computed.

This comment is added in the previous sentence when the features are introduced.

Line 216: Is this due to some physical processes?

At this point of the manuscript, the text only describes the morphology, to unbias the reader. The physical interpretation is done at the end when all pieces (dust, synchrotron, gas tracers,

transfer models) are together. Yes, we attribute it to the feeding material in the torus (this is mentioned in the abstract and final discussion).

Lines 222-228: From the grey contours, also in the eastern region seems that the synchrotron emission shows an arc feature and it seems cospatial just in the first parsec.

Yes, we added it to the text.

Lines 228-230: Consider overlaying MATISSE N-band contours in Fig. 2h.

We tried it but it makes the plot very busy. After considering this, we left this figure untouched.

Lines 235-237: Is this due to some physical processes?

As mentioned above, this part of the manuscript only describes morphology. The physical interpretation is done at the end when all tracers and models are put it together.

Lines 257-258: Why the extended emission is just toward the base of the North Arc and there is a lack of emission in the eastern region.

The H36alpha traces the hot ionized outflow which seems to be only present within the central pc. The galaxy may obscure the Eastern region. We do not get into this as it is discussed and analyzed in a lot of detail in Izumi+2023.

Lines 263: specify what the HCN(3-2) emission trace. Could be worth to note the lack of emission in the centre ?

The critical density of the HCN(32) molecular gas was quoted to compare with that from the CI molecular gas emission. The lack of emission in the core is omitted because it is discussed in Izumi+2023, due to text limit constraints.

Lines 291-295: Why not citing also [CI]?

DONE

Line 308: In line 154 the PA is -64deg. Clarify how you compute this.

DONE. The -64 ° should be -70°. The text has been modified. The -70deg is estimated as the long-axis of the 2D Gaussina fit to the extended emission as shown in Fig. 8.

Line 310: How this dust temperature compares with the temperature reported in Lines 193-195?

DONE

Line 334: Which observations you are using for your photometric SED.

This is presented in the Method section.

Lines 369-375: Rephrase avoiding repetition.

DONE

Lines 403-404: Explain also why it is not seen at 890 μm

DONE. This is answered in the previous section.

Line 411: The used aperture is 8pc or 12.3pc as indicate in App.9 (Fig10)?

DONE

Line 412: 'North Arc'

DONE

Lines 420-421: Truncated phrase (probably refers to Fig. 3?)

DONE

Lines 441-442: References not linked.

DONE. At the time of submission, these papers were not publicly available. Only Lopez-Rodriguez+2025a is available in ArXiv.

Lines 558: Say why you align with the 1200 μm ALMA.

DONE

Lines 573: fig. -> Fig.

DONE

Lines 576: "Winds"?

DONE

Lines 587-592: Say which instrument you used to compute the contribution of the lines.

DONE

Line 603: Specify which has lower resolution since you used many observations.

DONE

Fig.8: Optimize space in the figure using just one colorbar, put logarithm in the label and use arcsec instead of pixel as in other Figs.

As an appendix and supplementary figure, this figure only shows the fitting and the residuals. We would leave this image as presented.

Reviewer #3 (Remarks to the Author):

I believe this work represents an important step forward in the study of active galactic nuclei and their environments. I recommend publication following a few relatively minor changes and/or clarifications.

#####

Major Comments

#####

In the discussion about directly illuminated PA~64 deg dust in the NLR, the authors cite Stalevski et al. (2017) in the context of radio jet illumination. However in that work (and Stalevski et al. 2019), the illumination is instead attributed to the warped accretion disk. If the authors prefer the jet-based interpretation, it would be helpful to expand the discussion and clearly justify this deviation from earlier interpretations.. Moreover, it may be helpful to note that the maser disk is thought to be warped (Greenhill 2003), causing anisotropic illumination of the dusty structures. This is explored in Jud et al. (2016) and used as the reasoning for the preferential illumination of one edge of the outflow cone in Stalevski et al. (2017, 2019).

Lines 264-267 mention the anisotropic (east-west elongated structures) illumination by Stalevski+2017, and we attributed our measured >5 pc extended emission along that direction to this interpretation (lines 316-318, 357-360). The text now includes a mention to the warped maser disk (lines 267-269)

In fitting the SED with RT models, the authors assume the disk to be perfectly edge-on. Previous authors, such as Stalevski et al. (2019) required a slight inclination (~85 deg) in order to explain bright MIR emission (rather than absorption) along the torus PA. Do the authors expect the results of the SED fitting (and which sets of models work best) change when using the previous value of ~85 deg instead of 90 deg?

The change of 5 deg. does not substantially modify the results of the torus models. As an example, HyperCAT (Nikutta+2021; <https://ui.adsabs.harvard.edu/abs/2021ApJ...919..136N/abstract>) shows the changes of elongation of image morphology as a function of inclination (their Fig. 6), and the morphological size as a function of inclination, number of clumps, and torus width parameter (their Fig., 5). There are no significant changes within 5 deg.

The authors claim that the torus-only models are statistically preferred to describe the 1-1000 micron SED. Table 2, however, includes many chi2 values and in many cases the torus-only models are not preferred (ie, exhibiting the smallest chi2). The authors should clarify which dataset (of the four) they use to make this claim – i.e., without JWST or with JWST and which aperture size.

The paragraph starting in line 376 discusses several preferred torus models and the SED used to obtain these results. In addition, lines 453-456 discuss that the torus+wind model can account for the extended component.

The choice to fit a pure blackbody rather than a graybody or a blackbody with foreground absorption is not fully justified. It may be helpful to clarify the wavelength range used for this fit, and to consider whether including a foreground extinction screen (e.g., $A_v \sim 28$ mag) would materially affect the inferred temperature. Additionally, typical ISM extinction will change the M-band fluxes quite a bit relative to the L-band (e.g., Fig 3; Schartmann et al. 2005) even relatively far from the silicate feature at ~ 10 micron. This changing MIR color will possibly impact a fitted temperature result (only quantitatively, and the qualitative results of the paper will remain).

Thanks for catching this point. Using a blackbody with foreground absorption (to avoid the assumption of the emissivity index parameter if using a graybody), the characteristic temperatures change from 450 K to 532 K, and 730 K to 1073 K for the extended component and the North arc, respectively. These numbers are now included in the text, which only changes it quantitatively, while the results are still the same.

```
#####  
Minor Comments  
#####
```

It is unclear where the uncertainty (and therefore significance of features) on the final images comes from. The SQUEEZE results are from an MCMC, meaning that a large number of images are created for each bootstrap sampling of the uv weights. The authors mention recovering 100 different images and averaging those, but are each of those 100 images already an average of the SQUEEZE MCMC sampling?

To characterize the signal-to-noise ratio (SNR) of the images, we recovered 100 images per data cube with different samples of the observed uv-plane. For this procedure, we randomly sampled the uv frequencies of the interferometer by changing their weights; at the same time, we kept the total number of uv points constant. Each image is an MCMC of a random sample of the uv plane with different weights. Finally, we averaged the 100 different reconstructions per wavelength to construct the final image (Lines 577-580).

The authors mention that the North Arc is filtered out in the MATISSE images. Rather than listing the smallest scales that the MATISSE observations could attain, it might be useful to list

the largest scales the MATISSE observations were sensitive to. This may aid in interpretation of the apparent M-band PA discrepancies between the two datasets.

The maximum recoverable angular scale in L and N bands is included.

Could the authors please explain why they are displaying/comparing to the 10.5 micron image from Isbell et al. (2022)? The 10.5 micron image is still within the rather deep, broad silicate feature they report, and so it is not a continuum image. The 11.3 or 12.0 micron image would better represent the continuum. If this image was selected to match the 10 micron image of e.g., Stalevski et al. (2017), the authors could instead also use the 11.9 um image from that paper.

The MATISSE image was changed to the publicly available 12.0 um image from Isbell+2022 (<https://cdsarc.cds.unistra.fr/viz-bin/cat/J/ApJ/910/104>).

The flux densities are apparently negative in Fig 8. Are they meant to represent fractions of the peak in logarithmic spacing?

The fluxes are in log-scale. The new caption mentions that.

Line 421 appears to reference Figure 6, but based on the context, this may have been intended to refer to Figure 3 or 10.

Changed to figure 3.

It is unclear to me at which position the fluxes/temperatures of the northern arc are being measured. Adding apertures to Fig 8 would clarify the location of the temperature estimate (~720 K) and the scale of the structures being measured.

The apertures are shown in Figure 8.

The authors discuss the discrepancy between most of the models and the SED at ~5um (shown in e.g. Fig 10) and state that CAT3D-WIND can help describe this. Fig. 10 shows that this discrepancy becomes less significant with larger aperture sizes. Does this indicate that the feature is arising from smaller scales?

Indeed, to explain the large aperture fluxes, the torus+wind seems to include an additional hot dust component missing in the torus-only models. Note that the large aperture includes the flux from the 'North Arc', which can explain this result. We included this result in lines 453-456, we discuss that the torus+wind model can account for the extended component.

It may be worth pointing out more explicitly that while the chi2 comparisons to the various models preferred the Torus alone, the Hypercat comparisons and observed morphology point almost directly to a disk+wind scenario. In the works of Gonzalez-Martin et al. (2019) and Isbell et al. (2021), the Nenkova2008 models are shown to cover a large parameter space in NIR/MIR colors – larger than the other torus(+wind) models analyzed. Could this play a role in the

statistical preference for torus-only models via χ^2 in this paper, when the morphology and Hypercat comparisons indicate otherwise?

HyperCAT uses the best-fit of the clumpy torus model (torus-only), the morphological comparison with dust continuum and outflows, and dust temperatures point to the extension to be dominated by dust on the funnel of the torus. Only when the large aperture takes into account the 'North Arc' emission, the torus+wind may be able to explain the SED, large-scale morphology, including the 'North Arc' dusty and molecular outflow. The conclusions of this work are based on the accumulated evidence not only on the best fit model to the SED. Also note that the resolved torus by ALMA is a significant constraint, and only the torus-only models are close to it, mainly because the torus+wind models underpredict the torus size and mass by putting too much dust along the polar directions.

Dear Anonymous Referees,

Thank you for the comments on the revised version. We have implemented all your suggested changes and addressed your comments point-by-point below. The new version has the implemented revisions in bold.

Reviewer #1 (Remarks to the Author):

I appreciate the authors for thoroughly revising the manuscript.

I found that the manuscript is almost satisfactory but I have a remaining question regarding the point 2 in my original draft.

>(2) The hypothesized torus is not observed even with the 1.2 mm wavelength by ALMA. Is it possible to put constraints on the properties of the dust (total mass, size, and temperature) in the torus?

>The dusty torus is observed with ALMA at 700 μm (Figure 2-f), and the molecular torus is observed with ALMA at CO(6-5) (Figure 2-j) and HCN(3-2) (Figure 2-l). Our HyperCAT models at 700 μm reproduce the thermal emission of the dusty torus as shown in Figure 3. Note that the 1.2 mm AMA observations are dominated by the radio synchrotron emission from the radio jet, which is shown in Figure 2-h and the SED in Figure 9.

I cannot find an answer to my question about the constraints on dust properties.

Is it impossible to put a constraint on the dust mass and size in the torus?

Yes, it is possible to put constraints on the mass and size of the torus. We apologize for overlooking this part of the comment and thank you for pointing it out. The size of the torus is $5 \times 3 \text{ pc}^2$ and the median dust mass is $\sim 4 \times 10^3 M_{\text{sun}}$ for the clumpy torus model (preferred model from the fitting). We have included this information in the main body of the manuscript and added a more detailed explanation with comparisons in the methods section 'torus model', due to space constraints.

Reviewer # 2 (Remarks to the Author):

The author addressed all the comments.

Few minor comments:

20: ,, and units.

The units were removed. They already are in arcseconds (").

295 trace -> traces

Done.

Reviewer #3 (Remarks to the Author):

The authors have satisfactorily answered my major concerns. In particular, the inclusion of the apertures in Fig. 8 and the description of the image reconstruction enhance clarity. I thank the authors for their detailed responses.

We are glad it helped.

The authors are correct in pointing out that after the inclusion of the foreground dust screen, the temperatures are qualitatively the same. Outside the scope of this paper, the much higher inferred North Arc temperature could have interesting implications for dust density/clumpiness in the so-called polar wind.

Agree with this comment. Interesting follow-up project!

Regarding my previous comments about the torus-only vs disk+wind model as inferred from the chi2 comparisons and to HyperCAT: I did not fully appreciate the constraints that the ALMA torus provided to the interpretation. With the revised text and the authors' responses, it is now clearer to me how the SED in combination with the morphological constraints led to the authors' interpretations. In particular, the addition of Figure 11 helps illustrate the regimes in which the "torus" dominates or in which the "wind" dominates.

We are glad it helped.